# WaveFormer: Leveraging Wavelet Transform for Multi-Scale Token Interactions in Hierarchical Transformers

## Abstract

Recent transformer models have achieved state-of-the-art performance for visual tasks involving high-dimensional data like 3D volumetric medical image segmentation. Hierarchical transformers (e.g. Swin Transformers) circumvent the computational challenge of self-attention in high-dimensional data through shifted window approach to learn token relations within progressively overlapping local regions, thus expanding receptive field across layers while limiting token attention span in each layer within predefined windows. In this work, we introduce a novel learning paradigm that captures token relations through progressive summarization of features. We leverage the compaction capability of discrete wavelet transform (DWT) on high-dimensional features and learn token relation in multi-scale approximation coefficients obtained from DWT. This approach enables efficient representation of fine-grained local to coarse global contexts within each layer of the network. Furthermore, computing self-attention on the DWT transformed features significantly reduces the computational complexity, effectively addressing the challenges posed by high-dimensional data in vision transformers. Our network competes favorably with current SOTA transformers (e.g. SwinUNETR) using three challenging public datasets on volumetric medical imaging: (1) MICCAI Challenge 2021 FLARE, (2) MICCAI Challenge 2019 KiTS, and (3) MICCAI Challenge 2022 AMOS. Our DWT-based transformer termed as WaveFormer consistently outperforms Swin-UNETR with improvement from 0.929 to 0.938 Dice (FLARE2021) and 0.880 to 0.900 Dice (AMOS2022). The source code and pretrained models will be made available in the full paper submission.

## 1 Introduction

The Vision Transformer (ViT) architecture Dosovitskiy et al. (2020) has proven to be highly effective for visual recognition tasks due to its ability to model long-range relationships across non-overlapping image patches or tokens. However, ViT comes with significant computational costs, as its self-attention mechanism scales quadratically with input size. In addition, ViT generates low-resolution single-scale output features that are unsuitable for downstream tasks that require fine-grained analysis of high-resolution feature maps and global context understanding (Beal et al., 2020; Fang et al., 2021; Xie et al., 2021; Zheng et al., 2021). These challenges are especially significant for high-dimensional inputs such as 3D volumetric scans. Hierarchical backbones Wang et al. (2021); Liu et al. (2021) offer a solution by reducing computational complexity through local window attention applied to progressively smaller feature maps. While this alleviates some of the computational burdens, it introduces a new limitation. The effective receptive field (ERF) becomes constrained within each layer, even after techniques like neighborhood pooling Yang et al. (2021) and shifted windows Liu et al. (2021) are applied. These methods attempt to expand the receptive field in subsequent layers by gradually exposing tokens to previously unseen tokens, but the restriction within the individual layers remain.

Recent studies demonstrate that self-attention mechanisms in ViTs exhibit characteristics analogous to a low-pass filter, as in, low-frequency components are crucial for the performance of ViT models Bai et al. (2022); Wang et al. (2022b); Park & Kim (2022); Rao et al. (2021); Wang et al. (2020a; 2022a). In this work, we propose that it is feasible to achieve a multi-resolution feature

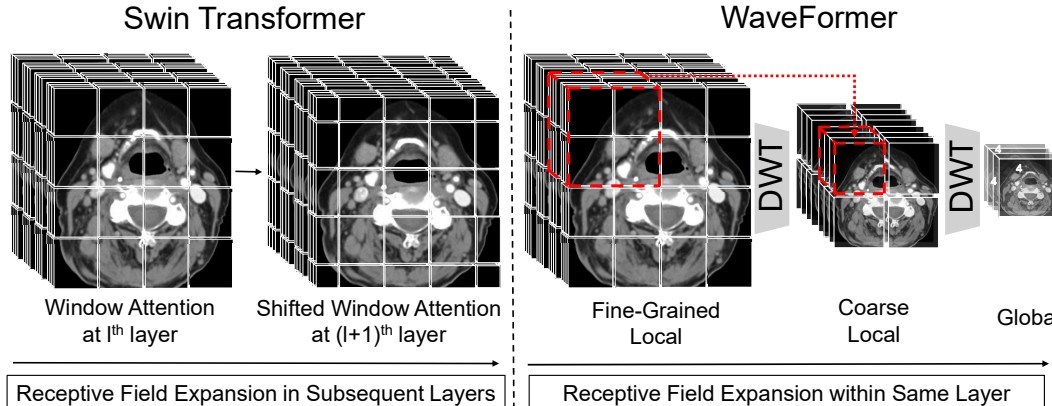

Figure 1: Comparison of token relation learning mechanism between Swin (left) and WaveFormer (right). Each finest volumetric cube (shown in white) represents the span of window self-attention ($4 \times 4 \times 4$). Swin expands the receptive field through the shifted window mechanism in subsequent layers. On the contrary, WaveFormer captures local and global relations in each layer on the multi-scale low-frequency approximations obtained using DWT. The window size is carefully configured to match the feature map length/width at the coarsest scale, thus leading to compute global attention; while allowing multi-granular local attention on other scales. The dashed red cube illustrates the summarization of features and resulting widening of receptive field through one level of DWT. For visual clarity, high-frequency coefficients from DWT are not shown.

representation with reduced computational overhead by exploiting the inherent frequency-domain properties of images. Our approach computes patch/token relationships across multiple scales of low-frequency sub-bands derived through Discrete Wavelet Transform (DWT). This methodology enables the model to capture multi-scale context at each network layer, providing an efficient mechanism for processing high-dimensional data such as 3D medical scans. This technique expands the effective receptive field beyond what conventional window attention methods can achieve, as illustrated in Fig. 1.

Specifically, we propose a novel wavelet-based transformer architecture that decomposes features using DWT and computes windowed attention on the low-frequency components. Different level of decomposition enables attention at different resolutions, which allows the model to capture and aggregate essential local and global context at each stage. By prioritizing these compact low-frequency approximations, our method reduces the computational burden associated with high-resolution image analysis while preserving essential multi-resolution context. We validate our approach in 3D volumetric segmentation benchmarks, including FLARE Ma et al. (2022), AMOS Ji et al. (2022) and KiTS Heller et al. (2020b), where our model achieves state-of-the-art (SOTA) mean dice score. Additionally, our model demonstrates competitive results on classification with ImageNet-1k Deng et al. (2009), highlighting its generalization ability across medical and natural image recognition tasks. Our contributions can be summarized as below:

- We introduce WaveFormer, a novel transformer architecture that processes low-frequency approximations of spatial images through DWT. This approach enables multi-resolution contextualization of visual elements, resulting in a significant expansion of the effective receptive field while maintaining superior computational efficiency compared to similarly sized models.

- Our model capitalizes on the high energy density present in low-frequency components, optimizing representation learning from natural and volumetric images. This novel integration of the discrete wavelet transform opens new pathways for efficiently processing large-scale visual data.

- Our extensive experiments demonstrate that WaveFormer surpasses state-of-the-art performance on 3D volumetric segmentation tasks, achieving superior mean dice scores on the FLARE, AMOS and KiTS test sets. Additionally, our model achieves competitive accuracy on ImageNet-1k for natural image classification, all while reducing FLOP counts compared to other models in its class.

## 2 RELATED WORKS

### 2.1 RECEPTIVE FIELD - COMPUTATION SPECTRUM

Vanilla ViT Dosovitskiy et al. (2020) enjoys global receptive field by processing an entire sample as patchified input tokens, incurring massive computational burden ($O(N^2)$). In contrast, Stand-alone Self-attention Ramachandran et al. (2019) reduces computation by attending within non-overlapping local windows, limiting the receptive field to the window size. Various approaches aim to balance the trade-off between computational cost and receptive field in transformer models. SWIN Liu et al. (2021) uses shifting windows between consecutive self-attention blocks for cross-window interaction, which adds complexity and limits global context. LinFormer Wang et al. (2020b) reduces computation via token projection, sacrificing fine-grained detail. Performer Choromanski et al. (2020) approximates attention with kernel methods, reducing computation to linear but yielding unreliable performance across tasks and modalities. Reformer Kitaev et al. (2020) hashes queries into buckets, risking sub-optimal grouping. Axial Attention Ho et al. (2019) processes 2D attention as sequential 1D attention, limiting global context capture. Longformer Zhang et al. (2021) and RegionViT Chen et al. (2021) focus on regional tokens but add complexity and limit global efficiency. Biformer Zhu et al. (2023) adapts to multi-scale contexts but has inconsistent performance. Focal AttentionYang et al. (2021) combines fine and coarse features but struggles with scalability. Dilated Attention Hassani & Shi (2022) takes adaptively spaced tokens which allow a larger receptive field at a low cost, but the resulting sparsity affects the attention granularity.

### 2.2 LEARNING IN FREQUENCY DOMAIN

Learning in the frequency domain has been explored in various tasks like image deblurring and image inpainting, often by learning directly from the frequency components, or as an assistive representation alongside the spatial domain Xu et al. (2020); Wang & Sun (2022); Gueguen et al. (2018); Bai et al. (2022); Zou et al. (2021); Suvorov et al. (2022); Ehrlich & Davis (2019). Some works have leveraged frequency for model compression Kong et al. (2023) and channel description Qin et al. (2021). Based on energy under low-frequency coefficients, Wang et al. (2022b) performs channel and token pruning to compress models. Yao et al. (2022) uses selective coefficient tokens for attention. However, such pruning or selective token shortlisting may cause information imbalance and redundancy. Additionally, the feature stacking and restoration in Yao et al. (2022) require extra layers, diminishing the computational benefits of the wavelet transform.

Compared to these works, our models' strength comes from integrating wavelet into a multi-path hierarchical architecture. Each branch in our attention block independently attends to features at different scales, capturing a broader range of patterns and scale invariance. Aggregating these branches helps contextualize multi-resolution object properties. Our in-depth analysis shows that such a multi-path network allows each path to develop distinct modeling abilities due to their differences in ERF.

## 3 WAVEFORMER: INTUITION

WaveFormer introduces a novel approach to hierarchical transformers by combining two key intuitions: learning on compact representations and achieving local-to-global receptive field coverage. The first notion leverages the properties of the Discrete Wavelet Transform (DWT) and Parseval's theorem to establish the significance of low-frequency approximations in the context of learning. This enables reduced computation while preserving essential global features. The second notion consolidates extraction of multi-resolution token relations by using multi-level DWT, which seamlessly models local and global dependencies. Together, these two intuitions form the foundation of our WaveFormer architecture, enabling efficient yet powerful token relation modeling.

### 3.1 LEARNING ON COMPACT REPRESENTATION

**Discrete Wavelet Transform:** The Discrete Wavelet Transform (DWT) decomposes a signal into coefficients that represent both spatial and frequency information at different scales. In contrast to the global nature of the Fourier Transform, DWT offers localized time-frequency analysis, making it

ideal for processing non-stationary signals, such as images. Given a 2D feature map $X \in \mathbb{R}^{C \times H \times W}$, DWT decomposes its spatial dimensions $(H, W)$ into an approximation coefficient $C_j$ and three detail coefficients $D_{j,k}$, representing horizontal ($k = 1$), vertical ($k = 2$), and diagonal ($k = 3$) orientations at each resolution level $j$.

Mathematically, the components from one-level DWT of $X$ can be expressed as:

$$C_1(c, h', w') = \sum_{h=1}^{H} \sum_{w=1}^{W} X(c, h, w) \cdot \phi_{h'}(h) \cdot \phi_{w'}(w), \tag{1}$$

$$D_{1,k}(c, h', w') = \sum_{h=1}^{H} \sum_{w=1}^{W} X(c, h, w) \cdot \psi^{(k)} h'(h) \cdot \psi^{(k)} w'(w), \tag{2}$$

where $\phi$ denotes the scaling (low-pass) function, $\psi^{(k)}$ denotes the wavelet (high-pass) functions for different orientations, and $(h', w')$ are the downsampled coordinates due to the subsampling operation in DWT.

By recursively applying DWT to the approximation coefficients $C_j$, we obtain a multi-level decomposition:

$$X(c, H, W) = C_J(c, h'', w'') + \sum_{j=1}^{J} \sum_k D_{j,k}(c, h_j, w_j), \tag{3}$$

where $J$ is the total number of decomposition levels, $h_j = H/2^j$, $w_j = W/2^j$, and $(h'', w'') = (H/2^J, W/2^J)$ represent the dimensions at the coarsest scale.

**Parseval's Theorem:** Parseval's theorem shows that the total energy of a time-varying signal $f(t)$ is preserved in its frequency domain representation $F(\omega)$, as expressed by Equation 4 Hassanzadeh & Shahrrava (2022).

$$\int_{-\infty}^{\infty} |f(t)|^2 \, dt = \frac{1}{2\pi} \int_{-\infty}^{\infty} |F(\omega)|^2 \, d\omega \tag{4}$$

When most of a signal's energy is concentrated in the low-frequency coefficients, transformations can be efficiently approximated by focusing on these components, significantly reducing computation. It has been observed in the literature Wang et al. (2022b); Park & Kim (2022) that in large-scale transformer models, features used for computing token relations in self-attention mechanisms predominantly reside in the low-frequency domain.

Using the orthonormality property of the wavelet transformations, it can be shown that energy of $X$ follows Parseval's theorem in the wavelet domain as mentioned in equation 5. Detailed derivation is provided in appendix A.1.

$$\|X\|^2 = \sum_{c=1}^{C} \sum_{h=1}^{H} \sum_{w=1}^{W} \left( |C_J(c, h'', w'')|^2 + \sum_{j=1}^{J} \sum_k |D_{j,k}(c, h_j, w_j)|^2 \right) \tag{5}$$

In conclusion, DWT offers three primary features that motivates our architecture:

- **Energy Compaction**: As feature energy in transformer networks is mostly aligned towards the low-frequency spectrum, DWT enables the concentration of the signal energy into a few approximation coefficients at the coarsest scale (follows from Parseval's Theorem).

- **Computational Efficiency**: By operating on wavelet coefficients at coarser scales, we reduce the computational burden without significant loss of important information.

- **Multi-Resolution Representation**: DWT provides a method for hierarchical decomposition of data. In spatial context, shallower level of decomposition represents local details as deeper levels tend to represent global structures. This enables another speculation for feature extraction at multiple scales, as discussed in Section 3.2 in more detail.

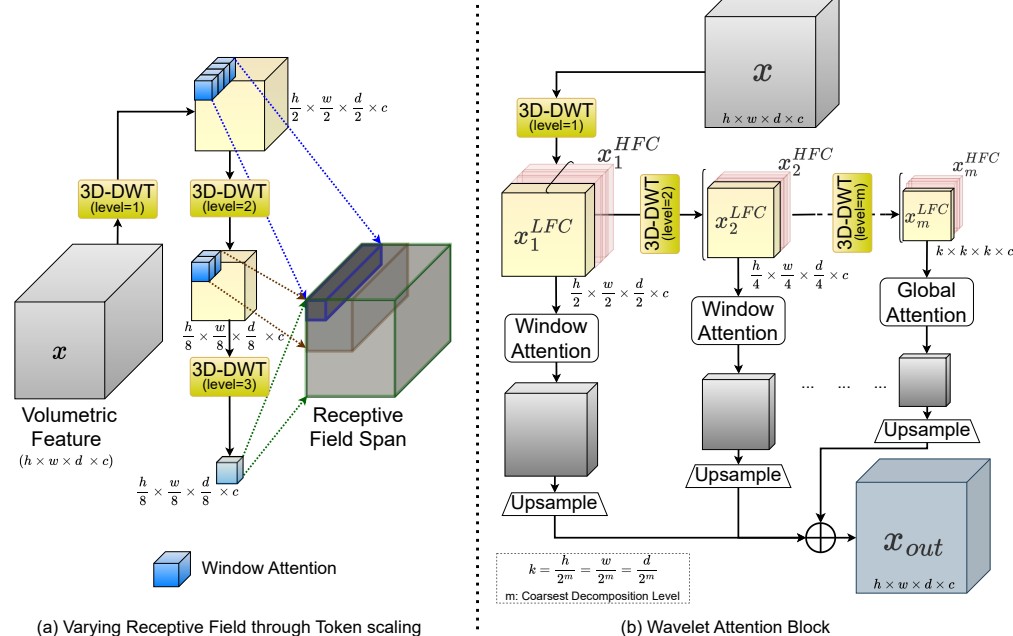

(a) Varying Receptive Field through Token scaling · (b) Wavelet Attention Block

Figure 2: **(a)** An illustration of how window attention in multiple resolutions enables capturing relationships that span multi-scale receptive fields in our network. The coarsest scale approximation ($\frac{h}{8} \times \frac{w}{8} \times \frac{d}{8}$) obtained from DWT is utilized to capture global context. Alongside this, the local relationship is captured through window attention from the intermediate approximations, where the window size is the same as the spatial shape of the coarsest scale feature. **b**: illustrates our wavelet-attention block. Input tokens are decomposed into low-frequency coefficients (LFC, shown as yellow cubes) and high-frequency (detail) coefficients (HFC, shown as red) of $M = 1, 2, ..., m$ scales using 3D-DWT. At each scale, window attention ($k \times k \times k$) is applied on the LFCs where $k = \frac{h}{2^m} = \frac{w}{2^m} = \frac{d}{2^m}$ i.e. the side of coarsest scale approximations $x_m^{LFC}$. This leads to capturing global attention from $x_m^{LFC}$ and multi-granular local attentions on $x_i^{LFC}; i = [1, m-1]$. Low-energy-density HFCs are omitted in our network.

## 3.2 LOCAL-TO-GLOBAL RECEPTIVE FIELD COVERAGE

As mentioned above, our encoder network computes token relations on the compact approximation coefficients obtained from the Discrete Wavelet Transform (DWT). Figure 4a illustrates DWT transformation on the input feature $x$, which is decomposed into multi-level low-frequency approximations. At the coarsest level, global attention is applied, enabling the capturing of holistic relationships among tokens. On other levels, the token relationship is computed locally using fixed-size window attention, where the window has the same shape as the spatial dimension of the coarsest-level feature. In this way, the attention mechanism efficiently captures multi-granular relationships spanning from local to global receptive fields as depicted in Figure 4a. This surpasses the limitation of window attention and introduces a mechanism that learns token relation through multi-level summarization of the input feature with low computational cost. Such a straightforward and effective approach to capturing token relationships at multiple resolutions has inspired us to develop a wavelet-decomposition-based transformer network.

In the context of our WaveFormer architecture, we apply DWT to the input feature map $x$ to obtain a set of approximation coefficients $C_j$ at multiple scales. Using these low-resolution wavelet coefficients $C_j$, we capture global and local dependencies with reduced computation by applying self-attention on the compact representations, enhancing efficiency without sacrificing accuracy.

## 4 WAVEFORMER: NETWORK ARCHITECTURE

WaveFormer, a hierarchical transformer, comprises multi-resolution window attention in compressed feature space. This enables the learning of token relations from high-dimensional data like

medical computed tomography (CT) scans with reasonably less computational overhead. Multi-resolution features are obained in the encoder by applying attention on wavelet-approximated features. A convolution-based decoder network is used for downstream tasks which receives multi-stage encoder outputs via convolutional skip connections. Figure 3 illustrates the complete architecture of WaveFormer. In the following subsections, we describe the details of the encoder and decoder.

## 4.1 ENCODER: WAVELET-TRANSFORMATION BASED TOKEN RELATION

Random sub-volumes $S_i \in \mathbb{R}^{H \times W \times D \times P}$ are extracted from a set of 3D Image Volumes $V_i = X_i, Y_{i \ i=1,2,\ldots,L}$ and passed as input to the encoder network. A simple convolutional embedding is applied to the input to create 3D tokens of dimensions $\frac{H}{2} \times \frac{W}{2} \times \frac{D}{2}$ that is projected to a $C = 48$ dimensional space. Following Hatamizadeh et al. (2021), this embedding is passed through 4 encoder stages where in each stage we have 2 wavelet-attention blocks (i.e. $L = 8$ total layers) as depicted in Figure 3. Patch embedding is applied after each stage (except the last one) to obtain hierarchical feature. After each stage we obtain feature map $F_i$ of size $\frac{H}{2^i} \times \frac{W}{2^i} \times \frac{D}{2^i} \times 2^{i-1}C$ at stage $i$ where $i \in \{1,2,3,4\}$.

**Wavelet Attention Block.** Instead of calculating token relations on the original patch embedding feature $X \in \mathbb{R}^{h \times w \times d \times c}$, where $h$, $w$, $d$ and $c$ represent the height, width, depth and dimension at stage $i$, self-attention mechanism is applied to the multi-scale ($M = 1, 2..., m$ scales) low-frequency approximation coefficients of $X$ obtained by the discrete wavelet transform (DWT), as depicted in Figure 4b. On the coarsest $m^{th}$ scale, coarse global relation is captured through global attention while in other scales, window ($k \times k \times k$) attention is applied to capture multi-granular local information. For simplicity, we used $k = \frac{h}{2^m} = \frac{w}{2^m} = \frac{w}{2^m}$ i.e. the window size is same as the coarsest scale feature map. This mechanism effectively enables relation capturing across various receptive fields without the need of dynamic window-size or window shifting and further parameterization.

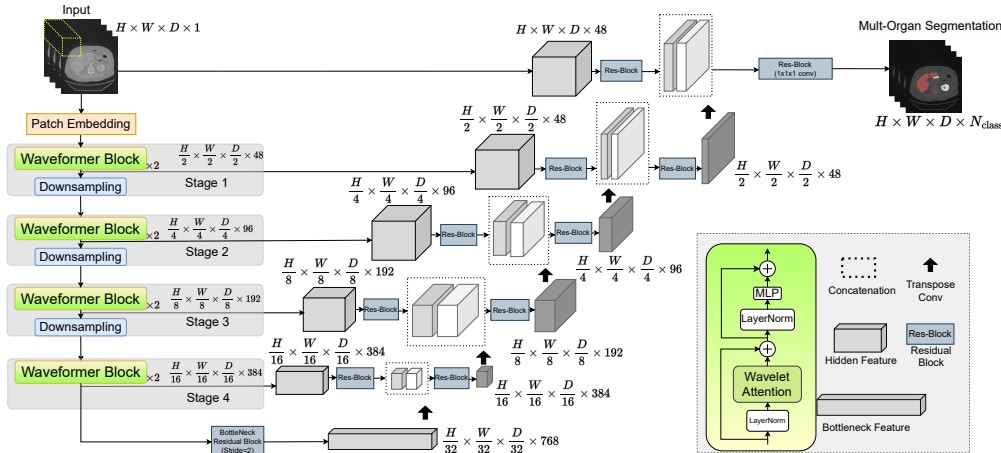

Figure 3: Model Architecture for our proposed WaveFormer network. 3D patch embedding is generated with Conv3D and passed through 4 stages of operation. In each stage, Waveformer block extracts multi-resolution salient features in depth-wise manner, and a following downsampling block mixes and enriches context across channels. For segmentation, features from each stage of encoder are collected through skip connection and final segmentation output is formed through progressive reconstruction.

## 4.2 DECODER FOR DOWNSTREAM TASK

For the downstream segmentation task, we follow the similar decoder architecture from Lee et al. (2022); Hatamizadeh et al. (2021) that comprises a "U-shaped" network overall. Multi-scale output from different stages of the network is connected to the corresponding decoder layer via a skip connection. First, the output feature from each stage $I(i \in 1, 2, 3, 4)$ is passed through a residual block comprised of two post-normalized $3 \times 3 \times 3$ convolutional layers with instance normalization. This stabilizes further propagation of the feature. Note that the feature from stage 4 is also

passed through a bottleneck residual layer to produce the final encoded feature. The feature is then upsampled with a transpose convolution and concatenated with the previous stage features. The concatenated feature will further be passed through a residual block to output the final feature for that decoder layer (dark gray in Figure 4a). For final segmentation, the residual feature from input patch is concatenated with the upsampled feature from the previous decoder layer and passed through a residual block with $1 \times 1 \times 1$ convolutional layer with a softmax activation to predict the segmentation probabilities.

## 5 EXPERIMENTAL SETUP

### 5.1 DATASETS

We experiment on 4 publicly available datasets to validate our model. For volumetric segmentation, we utilize MICCAI 2021 FLARE Challenge dataset Ma et al. (2022), MICCAI 2022 AMOS Challenge dataset Ji et al. (2022) and MICCAI 2019 KiTS Challenge dataset Heller et al. (2020a). For classification, we use the widely adopted Imagenet-1K dataset Deng et al. (2009). Additional details about the datasets are presented in Appendix A.3.

### 5.2 IMPLEMENTATION DETAILS

Following Lee et al. (2022), the model is evaluated in two scenarios for volumetric medical image segmentation: 1) directly supervised training on FLARE2021 and KITS2019 datasets, and 2) transfer learning with FLARE pre-trained wights on AMOS 2022 dataset. More detailed information on datasets and splits is provided in Appendix A.3. We performed 5-fold cross-validation on both FLARE and KITS while using the best fold model trained on FLARE to finetune on AMOS. Training details are provided in Appendix A.4. We evaluate WaveFormer against the current volumetric transformer and ConvNet SOTA approaches for volumetric segmentation in a fully-supervised setting. The dice similarity coefficient is used as the evaluation metric.

We further train the model on the natural image dataset Imagenet-1k for visual recognition tasks to test the generalization capability of the representation encoded by the model. Training details on Imagenet-1k are provided in Appendix A.5.

Furthermore, we performed ablation studies to investigate the effect of different-level wavelet decomposition on the model's capability to learn different-scale organs.

## 6 RESULTS

### 6.1 EVALUATION ON FLARE2021

Table 1: Performance comparison on FLARE 2021 datasets.

| Methods | #Params | FLOPs | FLARE 2021 | | | | |
|---|---|---|---|---|---|---|---|
| | | | Spleen | Kidney | Liver | Pancreas | Mean |
| 3D U-Net Çiçek et al. (2016) | 4.81M | 135.9G | 0.911 | 0.962 | 0.905 | 0.789 | 0.892 |
| SegResNet Myronenko (2019) | 1.18M | 15.6G | 0.963 | 0.934 | 0.965 | 0.745 | 0.902 |
| RAP-Net Lee et al. (2021) | 38.2M | 101.2G | 0.946 | 0.967 | 0.940 | 0.799 | 0.913 |
| nn-UNet Isensee et al. (2021) | 31.2M | 743.3G | 0.971 | 0.966 | 0.976 | 0.792 | 0.926 |
| TransBTS Wenxuan et al. (2021) | 31.6M | 110.4G | 0.964 | 0.959 | 0.974 | 0.711 | 0.902 |
| UNETR Hatamizadeh et al. (2022) | 92.8M | 82.6G | 0.927 | 0.947 | 0.960 | 0.710 | 0.886 |
| nnFormer Zhou et al. (2021) | 149.3M | 240.2G | 0.960 | **0.975** | 0.977 | 0.717 | 0.908 |
| SwinUNETR Hatamizadeh et al. (2021) | 62.2M | 328.4G | 0.979 | 0.965 | 0.980 | 0.788 | 0.929 |
| 3D UX-Net Lee et al. (2022) | 53.0M | 639.4G | 0.981 | 0.969 | **0.982** | 0.801 | 0.934 |
| **WaveFormer (ours)** | 52M | **326.56G** | **0.982** | 0.969 | 0.981 | **0.828** | **0.941*** |

The performance of our proposed WaveFormer model is compared against SOTA approaches for FLARE segmentation in Table 1. With the wavelet-decomposition-based multi-resolution attention transformer as the encoder backbone, WaveFormer significantly improves Dice scores on the FLARE2021 dataset. Specifically, WaveFormer outperforms competing models like TransBTS, UNETR, nnFormer, and SwinUNETR and achieves higher overall mean Dice scores (from 0.934 in 3D

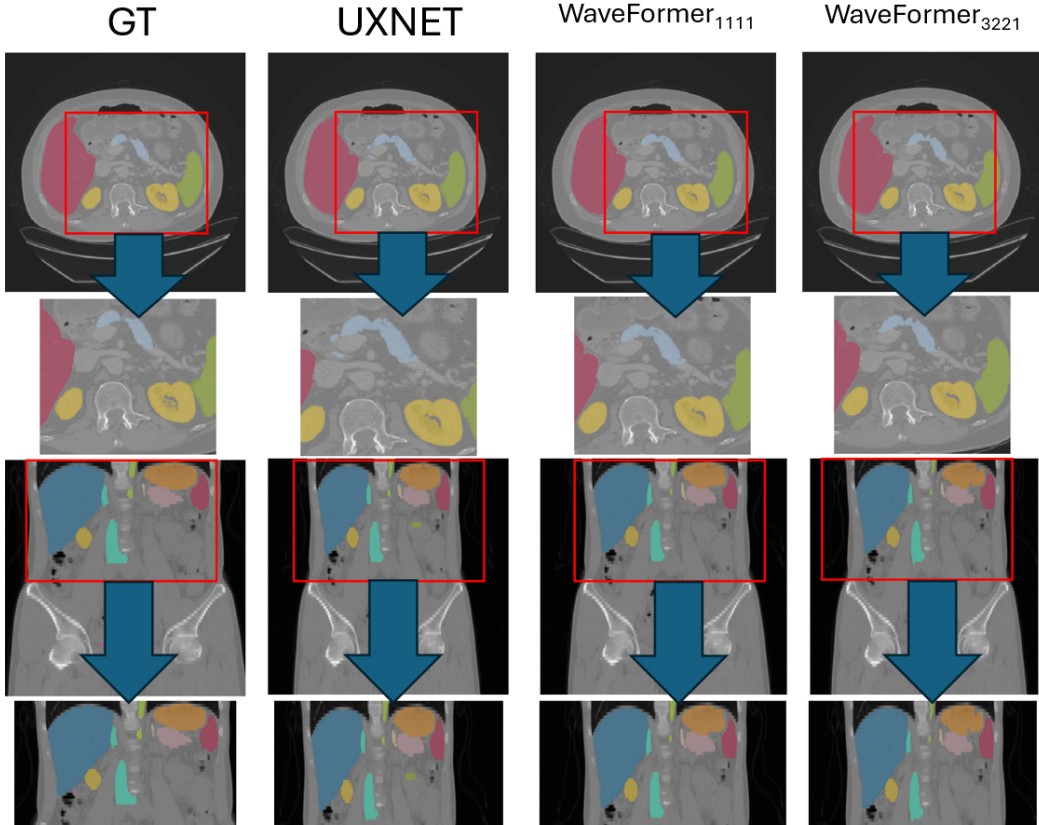

Figure 4: Qualitative representations of tissues and multi-organ segmentation across FLAIR2021 & AMOS2021 public datasets. Boxed are further zoomed in and visualize the significant differences in segmentation quality. WaveFormer shows the best segmentation quality compared to the ground-truth.

Table 2: Comparison of Finetuning performance with transformer SOTA approaches on the AMOS 2021 testing dataset.(*: $p < 0.01$, with Wilcoxon signed-rank test to all SOTA approaches)

| Methods | Spleen | R. Kid | L. Kid | Gall. | Eso. | Liver | Stom. | Aorta | IVC | Panc. | RAG | LAG | Duo. | Blad. | Pros. | Avg |
|---|---|---|---|---|---|---|---|---|---|---|---|---|---|---|---|---|
| nn-UNet | 0.965 | 0.959 | 0.951 | 0.889 | 0.820 | 0.980 | 0.890 | 0.948 | 0.901 | 0.821 | 0.785 | 0.739 | 0.806 | 0.869 | 0.839 | 0.878 |
| TransBTS | 0.885 | 0.931 | 0.916 | 0.817 | 0.744 | 0.969 | 0.837 | 0.914 | 0.855 | 0.724 | 0.630 | 0.566 | 0.704 | 0.741 | 0.650 | 0.792 |
| UNETR | 0.926 | 0.936 | 0.918 | 0.785 | 0.702 | 0.969 | 0.788 | 0.893 | 0.828 | 0.732 | 0.717 | 0.554 | 0.658 | 0.683 | 0.722 | 0.762 |
| nnFormer | 0.935 | 0.904 | 0.887 | 0.836 | 0.712 | 0.964 | 0.798 | 0.901 | 0.821 | 0.734 | 0.665 | 0.587 | 0.641 | 0.744 | 0.714 | 0.790 |
| SwinUNETR | 0.959 | 0.960 | 0.949 | 0.894 | 0.827 | 0.979 | 0.899 | 0.944 | 0.899 | 0.828 | 0.791 | 0.745 | 0.817 | 0.875 | 0.841 | 0.880 |
| 3D UX-Net | 0.970 | 0.967 | **0.961** | 0.923 | 0.832 | **0.984** | 0.920 | 0.951 | 0.914 | **0.856** | **0.825** | 0.739 | **0.853** | 0.906 | 0.876 | 0.900* |
| WaveFormer (ours) | **0.974** | **0.967** | 0.960 | **0.925** | **0.872** | 0.983 | **0.926** | **0.954** | **0.914** | 0.846 | 0.822 | **0.782** | 0.850 | **0.910** | **0.885** | **0.910*** |

UX-Net to 0.941 in Wavelet) with fewer parameters and lower FLOPs compared to 3D UX-Net. Notably, WaveFormer maintains SOTA performance with almost half the computational cost (FLOPs) of 3D UX-Net ($\approx 50\%$ decrease, from $639.4G$ to $326.56G$). Apart from the quantitative representations, Figure **??** further shows that the morphology of organs and tissues are well preserved in our model's prediction compared to the ground truth.

## 6.2 TRANSFER LEARNING WITH AMOS

Following Lee et al. (2022), we further investigate the transfer learning capability of our Wave-Former on the AMOS dataset. The finetuning performance of WaveFormer outperforms the SOTA large kernel convolution network Lee et al. (2022) by 1% and the transformer network Hatamizadeh et al. (2021) by 3%. Also, the qualitative representation Figure **??** shows that our model performs significantly better at maintaining edge clarity, especially in challenging dense segmentation scenarios, highlighting its effectiveness compared to other methods.

Table 3: Comparison of Models based on Accuracy, Flops, and Parameters on Imagenet-1K

| Model | Accuracy | Flops | Params |
|---|---|---|---|
| DeIT (Global) | 79.90% | 4.6G | 22.1M |
| PVT (Global) | 79.80% | 3.8G | 24.5M |
| RegionViT (Window) | 83.30% | 5.7G | 31.3M |
| Focal (Window) | 82.2% | 4.9G | 29.1M |
| Swin (Window) | 81.3% | 4.5G | 29M |
| **WaveFormer** | **80.9%** | **3.7G** | **28.5M** |

## 6.3 VISUAL RECOGNITION ON IMAGENET-1K

We further investigate the generalization capability of our proposed encoder by evaluating it on the visual recognition benchmark in the natural image domain. WaveFormer performs favorably in The performance of the proposed **WaveFormer** model was evaluated on the image classification task against several state-of-the-art transformer-based approaches, including both global and window-based models, as shown in Table 3. WaveFormer achieves a favorable performance with the lowest FLOPs and parameter count among the window-based models ($\approx 22\%$ fewer FLOPs than Swin), incurring only a 0.4% drop in accuracy compared to Swin. WaveFormer offers more flexibility by incorporating wavelet blocks with negligible FLOPs increase, which makes it effective for multi-scale visual tasks. Furthermore, WaveFormer outperforms state-of-the-art global attention-based models at lower Flops, highlighting its lightweight yet effectiveness in capturing local features. Detailed comparisons can be found in the ablation.

## 6.4 ABLATION STUDIES

We study how different configuration of Wavelet Attention block contributes to the efficiency of WaveFormer. We leverage FLARE and ImageNet-1K datasets for experimenting on the contribution by different settings. For convenience, we name the variants of WaveFormer based on the branches a particular input feature is transformed with at stage 1, 2, 3, 4; respectively. As such,
**WaveFormer**$_{1111}$ consists of one branch in each attention block. In each stage, input feature is transformed to coarsest resolution so that it equals to the window-length of window attention.
**WaveFormer**$_{2211}$ consists of 2, 2, 1 and 1 branches in the attention blocks across stages 1-4. This design facilitates more fine-grained local details than above.
**WaveFormer**$_{3211}$ differs with the former on stage-1, enforcing a medium fine feature map that enforces an intermediate fine-to-coarse representation through window attention.
**WaveFormer**$_{3221}$ differs with the former only on stage-3, which imposes late stage fine-granularity to the aggregated attention output.

Table 4: Mean DICE scores for each organ and overall mean DICE for each model across all folds.

| Model | #Params | FLOPs | Spleen | Right Kidney | Liver | Pancreas | Overall Mean DICE |
|---|---|---|---|---|---|---|---|
| **WaveFormer**$_{1111}$ | 52.26M | 326.3G | 0.983 | 0.967 | 0.981 | 0.817 | 0.937 |
| **WaveFormer**$_{2211}$ | 52.26M | 326.59G | 0.982 | 0.965 | 0.981 | 0.826 | 0.938 |
| **WaveFormer**$_{3211}$ | 52.26M | 326.62G | 0.982 | 0.966 | 0.981 | 0.827 | 0.939 |
| **WaveFormer**$_{3221}$ | 52.26M | 327G | 0.982 | 0.969 | 0.981 | 0.828 | 0.941 |

**Waveformer Variants on ImageNet-1K:** Table 5 presents classification performance from different variants of our models. From WaveFormer$_{1111}$ to WaveFormer$_{2211}$, we show that increasing early-stage local token relations improves performance. Comparison between WaveFormer$_{3211}$ and WaveFormer$_{3221}$ shows increasing late-stage local details yields even further increase in accuracy.

**Feature decomposition with Pooling:** We considered max pooling as a downsampling alternative to DWT in our WaveFormer$_{1111}$ setting. Results in Table 5 clearly shows the superiority of low-frequency components from DWT in retaining more salient information during spatial reduction of feature maps.

Table 5: Mean Top-1 Accuracy on ImageNet-1K for WaveFormer variants

| Model | #Params | FLOPs | Top-1 Acc. |
|---|---|---|---|
| WaveFormer$_{1111}$ (MaxPool) | 28.5M | 3.7G | 80.794 |
| WaveFormer$_{1111}$ (DWT) | 28.5M | 3.7G | **80.884** |
| WaveFormer$_{2211}$ | 28.5M | 3.83G | 80.965 |
| WaveFormer$_{3211}$ | 28.54M | 3.82G | 80.966 |
| WaveFormer$_{3221}$ | 28.55M | 4.35G | **81.104** |

## 7 DISCUSSION & FUTURE WORKS

In this work, we proposed a frequency-level learning module as a general feature extractor and adapted it into a generic encoder-decoder architecture for volumetric segmentation. Our findings indicate that the process of learning from full-resolution feature maps can be effectively approximated by computing multi-resolution token relationships in the frequency domain with fewer computation. Two key factors contribute to WaveFormer's performance. First, the Discrete Wavelet Transform (DWT) enables selective retention of high-energy, low-frequency coefficients from 3D feature maps, which minimizes redundancy when processing pairwise token relations. Second, the reduction in spatial dimensions achieved by DWT facilitates attention across feature maps at different scales. The use of self-attention with constant-sized window captures local relationships at various granularities while also summarizing global relationships efficiently in a continuous token space.

In future work, we aim to further investigate optimal configurations for diverse datasets and tasks. This includes exploring the role of high-frequency, low-information density coefficients, which were omitted in the current implementation. Understanding how these high-frequency components contribute to the learning process could unlock new avenues for fine-tuning WaveFormer's architecture, potentially enhancing its utility across a broader range of vision applications.

## 8 CONCLUSION

In this study, we introduced WaveFormer, a transformer-based architecture designed for high-dimensional medical image segmentation. By utilizing a discrete wavelet transform-based self-attention mechanism, WaveFormer efficiently fuses local and global token relations, leading to superior segmentation performance on 3D volumetric datasets like FLARE2021 and AMOS2022. Our approach reduces computational overhead while outperforming traditional methods, setting a benchmark for future research in visual transformers.

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

## A APPENDIX

### A.1 PARSEVAL'S THEOREM FOR WAVELET

The wavelet transformation of function $f$ in the time domain can be expressed in the following way.

$$f(t) = \sum_k c_{J,k}\,\phi_{J,k}(t) + \sum_{j=1}^{J}\sum_k d_{j,k}\,\psi_{j,k}(t) \tag{6}$$

Here,

$\phi_{J,k}(t)$ are the scaling functions at the coarsest scale $J$, representing the low-frequency components of the signal.

$\psi_{j,k}(t)$ are the wavelet functions at different scales $j$ and positions $k$, representing the high-frequency components of the signal.

$c_{J,k}$ are the approximation coefficients that capture the overall shape of the signal.

$d_{j,k}$ are the detail coefficients that capture finer details at different scales.

The energy of the function f(t) is expressed as

$$\|f(t)\|^2 = \int_{-\infty}^{\infty} |f(t)|^2\, dt \tag{7}$$

Expanding the square,

$$\|f(t)\|^2 = \int_{-\infty}^{\infty} \left( \sum_k c_{J,k}\,\phi_{J,k}(t) + \sum_{j=1}^{J}\sum_k d_{j,k}\,\psi_{j,k}(t) \right)$$

$$\left( \sum_{k'} c_{J,k'}\,\phi_{J,k'}(t) + \sum_{j'=1}^{J}\sum_{k'} d_{j',k'}\,\psi_{j',k'}(t) \right) dt$$

$$= \int_{-\infty}^{\infty} \left( \sum_k c_{J,k}\,\phi_{J,k}(t) \cdot \sum_{k'} c_{J,k'}\,\phi_{J,k'}(t) \right.$$

$$+ \sum_k c_{J,k}\,\phi_{J,k}(t) \cdot \sum_{j'=1}^{J}\sum_{k'} d_{j',k'}\,\psi_{j',k'}(t)$$

$$+ \sum_{j=1}^{J}\sum_k d_{j,k}\,\psi_{j,k}(t) \cdot \sum_{k'} c_{J,k'}\,\phi_{J,k'}(t)$$

$$\left. + \sum_{j=1}^{J}\sum_k d_{j,k}\,\psi_{j,k}(t) \cdot \sum_{j'=1}^{J}\sum_{k'} d_{j',k'}\,\psi_{j',k'}(t) \right) dt \tag{8}$$

Here, the wavelet functions $\phi_{J,k}(t)$ and $\psi_{j,k}(t)$ are orthonormal. This implies

$$\int_{-\infty}^{\infty} \phi_{J,k}(t)\phi_{J,k'}(t)\, dt = \delta_{kk'}$$

$$\int_{-\infty}^{\infty} \psi_{j,k}(t)\psi_{j',k'}(t)\, dt = \delta_{jj'}\delta_{kk'}$$

$$\int_{-\infty}^{\infty} \phi_{J,k}(t)\psi_{j,k'}(t)\, dt = 0$$

Here, $\delta_{kk'}$ and $\delta_{jj'}$ are Kronecker deltas, which are 1 when the indices match and 0 otherwise.

Using orthonormality, the energy function in equation 8 reduces to,

$$\|f(t)\|^2 = \sum_k |c_{J,k}|^2 + \sum_{j=1}^{J}\sum_k |d_{j,k}|^2 \tag{9}$$

Equation 9 reflects Parseval's theorem for wavelet decomposition.

## A.2 Model Configuration

Table 6: Configuration of the model's decomposition level for each stage with output size

| Encoder | Output | Decomposition Levels | | | | |
|---|---|---|---|---|---|---|
| | | WaveFormer$_{1111}$ | WaveFormer$_{2211}$ | WaveFormer$_{3211}$ | WaveFormer$_{2221}$ | WaveFormer$_{3221}$ |
| **Stage 1** | $H/2 \times W/2 \times D/2$ | 3 | 1, 3 | 1, 2, 3 | 1, 3 | 1, 2, 3 |
| **Stage 2** | $H/4 \times W/4 \times D/4$ | 2 | 1, 2 | 1, 2 | 1, 2 | 1, 2 |
| **Stage 3** | $H/8 \times W/8 \times D/8$ | 1 | 1 | 1 | 0, 1 | 0, 1 |
| **Stage 4** | $H/16 \times W/16 \times D/16$ | 0 | 0 | 0 | 0 | 0 |

## A.3 Public Dataset Details

Table 7: Complete details of three public datasets

| Challenge | FLARE | KiTS | AMOS |
|---|---|---|---|
| **Imaging Modality** | Multi-Contrast CT | Arterial CT | Multi-Contrast CT |
| **Anatomical Region** | Abdomen | Kidney | Abdomen |
| **Sample Size** | 361 | 210 | 200 |
| **Anatomical Label** | Spleen, Kidney, Liver, Pancreas | Kidney, Tumor | Spleen, Left & Right Kidney, Gall Bladder, Esophagus, Liver, Stomach, Aorta, Inferior Vena Cava (IVC), Pancreas, Left & Right Adrenal Gland (AG), Duodenum |
| **Data Splits** | 5-Fold Cross-Validation Train: 272 / Validation: 69 / Test: 20 | 5-Fold Cross-Validation Train: 152 / Validation: 38 / Test: 20 | 1-Fold Train: 160 / Validation: 20 / Test: 20 |

## A.4 Medical Data Pre-processing and Model Training Setup

Table 8: Hyperparameters used in training and finetuning on three public datasets

| Hyperparameters | Direct Training | Finetuning |
|---|---|---|
| Encoder Stage | 4 | |
| Layer-wise Channel | 48, 96, 192, 384 | |
| Hidden Dimensions | 768 | |
| Patch Size | $96 \times 96 \times 96$ | |
| No. of Sub-volumes Cropped | 2 | 1 |
| Training Steps | 40000 | |
| Batch Size | 2 | 1 |
| AdamW $\epsilon$ | $1e-8$ | |
| AdamW $\beta$ | (0.9, 0.999) | |
| Peak Learning Rate | $1e-4$ | |
| Learning Rate Scheduler | ReduceLROnPlateau | N/A |
| Factor & Patience | 0.9, 10 | N/A |
| Dropout | X | |
| Weight Decay | 0.08 | |
| Data Augmentation | Intensity Shift, Rotation, Scaling | |
| Cropped Foreground | ✓ | |
| Intensity Offset | 0.1 | |
| Rotation Degree | $-30°$ to $+30°$ | |
| Scaling Factor | x: 0.1, y: 0.1, z: 0.1 | |

## A.5 Training on ImageNet-1k

We compare different approaches on the ImageNet-1k dataset, which comprises 1.28 million training images and 50K validation images from 1000 classes. For fair comparison, we follow the training recipes in Touvron et al. (2021); Wang et al. (2021); Yang et al. (2021). All models are trained from scratch for 300 epochs with a batch size of 1024 distributed across 4 NVIDIA A100 GPUs (batch size of 256 in each GPU). An initial learning rate of $5 \times 10^{-4}$, weight decay of 0.05 and 20 epochs of linear warm-up is used. AdamW optimizer Loshchilov (2017) is used with a cosine learning rate scheduler. We followed the same set of augmentation as in Liu et al. (2021). During training, we

crop images randomly to $224 \times 224$, while a center crop is used during evaluation on the validation set. We performed ImageNet training on the publicly available Nautilus hypercluster by National Reserch Platform.

## A.6 TABLE FOLD

Table 9: Performance comparison for different models and configurations.

| Fold | Spleen $\mu$ | Right Kidney $\mu$ | Liver $\mu$ | Pancreas $\mu$ | All $\mu$ |
|------|--------|--------------|-------|----------|------|
| **Model checking v2 wf 1111** | | | | | |
| 0 | 0.9789 | 0.9667 | 0.9827 | 0.7975 | 0.9314 |
| 1 | 0.9835 | 0.9676 | 0.9816 | 0.8167 | 0.9373 |
| 2 | 0.9803 | 0.9614 | 0.9812 | 0.8080 | 0.9327 |
| 3 | 0.9806 | 0.9690 | 0.9822 | 0.8369 | 0.9421 |
| 4 | 0.9825 | 0.9663 | 0.9816 | 0.8203 | 0.9377 |
| **Wavelet two branch wf 2211** | | | | | |
| 0 | 0.9819 | 0.9656 | 0.9816 | 0.8262 | 0.9388 |
| 1 | 0.9818 | 0.9659 | 0.9776 | 0.8187 | 0.9360 |
| 2 | 0.9780 | 0.9631 | 0.9759 | 0.8204 | 0.9343 |
| 3 | 0.9786 | 0.9700 | 0.9716 | 0.8156 | 0.9340 |
| 4 | 0.9822 | 0.9677 | 0.9821 | 0.8147 | 0.9367 |
| **Wavelet without split wf 3211** | | | | | |
| 0 | 0.9828 | 0.9664 | 0.9813 | 0.8276 | 0.9395 |
| 1 | 0.9784 | 0.9635 | 0.9800 | 0.8298 | 0.9379 |
| 2 | 0.9822 | 0.9652 | 0.9703 | 0.8184 | 0.9340 |
| 3 | 0.9810 | 0.9675 | 0.9815 | 0.8178 | 0.9369 |
| 4 | 0.9807 | 0.9654 | 0.9800 | 0.8202 | 0.9366 |
| **Wave_wo_split_v2_wf_3221** | | | | | |
| 0 | 0.9789 | 0.9654 | 0.9803 | 0.8149 | 0.9349 |
| 1 | 0.9820 | 0.9700 | 0.9778 | 0.8215 | 0.9378 |
| 2 | 0.9825 | 0.9683 | 0.9807 | 0.8281 | 0.9399 |
| 3 | 0.9807 | 0.9692 | 0.9828 | 0.8128 | 0.9364 |
| 4 | 0.9805 | 0.9683 | 0.9813 | 0.8184 | 0.9371 |

Table 10: Comparison of WaveFormer configurations on the segmentation performance of various organs. (*: $p < 0.01$, with Wilcoxon signed-rank test to all configurations)

| Configurations | Spleen | R. Kid | L. Kid | Gall. | Eso. | Liver | Stom. | Aorta | IVC | Panc. | RAG | LAG | Duo. | Blad. | Pros. | Avg |
|---|---|---|---|---|---|---|---|---|---|---|---|---|---|---|---|---|
| WaveFormer$_{1111}$ | 0.9740 | 0.9669 | 0.9604 | 0.9214 | 0.8812 | 0.9829 | 0.9336 | 0.9505 | 0.9123 | 0.8425 | 0.8245 | 0.7748 | 0.8640 | 0.8982 | 0.8633 | 0.9033 |
| WaveFormer$_{2211}$ | 0.9691 | 0.9672 | 0.9607 | 0.9244 | 0.8664 | 0.9833 | 0.9423 | 0.9521 | 0.9163 | 0.8385 | 0.8197 | 0.7867 | 0.8524 | 0.9086 | 0.8783 | 0.9043 |
| WaveFormer$_{3211}$ | 0.9734 | 0.9648 | 0.9612 | 0.9209 | 0.8619 | 0.9816 | 0.9340 | 0.9540 | 0.9108 | 0.8502 | 0.8003 | 0.7671 | 0.8519 | 0.8980 | 0.8412 | 0.8980 |
| WaveFormer$_{3221}$ | 0.9736 | 0.9672 | 0.9585 | 0.9246 | 0.8719 | 0.9831 | 0.9257 | 0.9544 | 0.9143 | 0.8459 | 0.8220 | 0.7817 | 0.8476 | 0.9098 | 0.8846 | 0.9043 |

