# OpenReview forum: "WaveFormer: Leveraging Wavelet Transformation for Multi-Scale Token Interactions in Hierarchical Transformers"
_ICLR.cc/2025/Conference — ICLR 2025 Conference Withdrawn Submission_

### Official Review · Reviewer_i19Y · 2024-10-31

**Soundness:** 2
**Presentation:** 1
**Contribution:** 2
**Rating:** 1
**Confidence:** 5

**Summary:**

Authors introduced a learning paradigm that captures token relations through progressive summarization of features. They leveraged the compaction capability of discrete wavelet transform on high-dimensional features and learn token relation in multi-scale approximation coefficients obtained from DWT. They incorporated this module into an architecture for volumetric medical image segmentation.

**Strengths:**

1. Literature reviews are very detailed.
2. Description of methods and results are thoughtful.

**Weaknesses:**

1. Insufficient experimental evaluation: authors only evaluated this model on three CT datasets, and these datasets have same annotated organs. Thus, current experimental results cannot demonstrate the performance and generalizability of the network. Evaluations on other organs and other 3D modalities are necessary, such as spin segmentation in CT, tumor segmentation in CT, tubular structure (vessel) segmentation in CT and segmentation tasks on MR imaging (Brain Tumor segmentation on MR and ACDC dataset).
2. Limited comparison: authors only compared their model with baselines which were proposed before 2022. Medical image segmentation is a very popular topic, and various volumetric models have been proposed between 2022 and 2024, such as [3][4][5][6][7][8]. These models have achieved superior performance, and some have low computational complexity. Thus, current comparison results are totally insufficient. We suggest to compare with MedNext, UNETR++, and VSmTrans.
3. Low novelty. Authors propose to utilize the Wave Transform to aggregate features, but other modules and operations are borrowed from others' works, including window attention and global attention. The overall network architecture was proposed in UNETR [1] and Swin UNETR [2], and has also been widely used [3][4][6].
4. Performance is not promising. Authors propose the WaveFormer for computational efficiency, but this network do not have a low number of parameters and FLOPs. In Table 1, it has more Params and FLOPs than 3D UNet, SegResNet, RAP-Net, nnU-Net, and TransBTS. In Table 3, accuracy of WaveFormer was lower than other compared results, 80.9% for WaveFormer compared with 83.3% for RegionViT, 82.2% for Focal and 81.3% for Swin. Params of WaveFormer is 28.5M, much higher than DeIT (22.1M) and PVT (24.5M). These results demonstrated that the performance of WaveFormer is not promising.
5. Poor presentation. (1) figures and captions are overlapped, (2) missing figure labels reference in Line 421 and 430 (3) only qualitative comparison with UXNet (4) wrong bold results. In Table 3, 80.9% was not the best, but was bold. Same with 28.7M. Some results which are not the best one were bold. (4) lack sufficient descriptions in figure and table captions.
6. Missing significant descriptions and experiments. Authors evaluated the WaveFormer on ImageNet-1K, but they didn't provide architectures that they used. If they proposed WaveFormer for volumetric medical image segmentation, they didn't design experiments to compare it with DeIT, PVT, RegionViT, Focal, and Swin on medical image segmentation tasks. If you propose the WaveFormer as a general Transformer architecture, they didn't design experiments in semantic segmentation. We suggest to compare WaveFormer with Focal and Swin on Medical Image segmentation by incorporating them into the same segmentation architecture.

Minors. (1) Lacking training and test time. (2) Inappropriate descriptions in Abstract: compare with current SOTA transformers, but transformers compared in this manuscript are not SOTA now in 2024 (e.g. SwinUNETR and UX Net). Thus, comparing WaveFormer with them cannot demonstrate its superiority. (3) lack p-values and standard deviation (4) No visual interpretation for Wave transform.

[1] Hatamizadeh, A., Tang, Y., Nath, V., Yang, D., Myronenko, A., Landman, B., ... & Xu, D. (2022). Unetr: Transformers for 3d medical image segmentation. In Proceedings of the IEEE/CVF winter conference on applications of computer vision (pp. 574-584).
[2] Hatamizadeh, A., Nath, V., Tang, Y., Yang, D., Roth, H. R., & Xu, D. (2021, September). Swin unetr: Swin transformers for semantic segmentation of brain tumors in mri images. In International MICCAI brainlesion workshop (pp. 272-284). Cham: Springer International Publishing.
[3] Liu, T., Bai, Q., Torigian, D. A., Tong, Y., & Udupa, J. K. (2024). VSmTrans: A hybrid paradigm integrating self-attention and convolution for 3D medical image segmentation. Medical Image Analysis, 98, 103295.
[4] Shen, Q., Zheng, B., Li, W., Shi, X., Luo, K., Yao, Y., ... & Wei, Q. (2024). MixUNETR: A U-shaped network based on W-MSA and depth-wise convolution with channel and spatial interactions for zonal prostate segmentation in MRI. Neural Networks, 106782.
[5] Roy, S., Koehler, G., Ulrich, C., Baumgartner, M., Petersen, J., Isensee, F., ... & Maier-Hein, K. H. (2023, October). Mednext: transformer-driven scaling of convnets for medical image segmentation. In International Conference on Medical Image Computing and Computer-Assisted Intervention (pp. 405-415). Cham: Springer Nature Switzerland.
[6] Shaker, A. M., Maaz, M., Rasheed, H., Khan, S., Yang, M. H., & Khan, F. S. (2024). UNETR++: delving into efficient and accurate 3D medical image segmentation. IEEE Transactions on Medical Imaging.
[7] Chen, J., Mei, J., Li, X., Lu, Y., Yu, Q., Wei, Q., ... & Zhou, Y. (2023). 3d transunet: Advancing medical image segmentation through vision transformers. arXiv preprint arXiv:2310.07781
[8] Azad, R., Niggemeier, L., Hüttemann, M., Kazerouni, A., Aghdam, E. K., Velichko, Y., ... & Merhof, D. (2024). Beyond self-attention: Deformable large kernel attention for medical image segmentation. In Proceedings of the IEEE/CVF Winter Conference on Applications of Computer Vision (pp. 1287-1297).

**Questions:**

1. Why didn't you compare your model with UX Net? UX Net didn't achieve a SOTA performance and it has a high computational complexity
2. Why didn't you evaluate the model on MR images and other heterogeneous organs and tumors to demonstrate the generalizability of the network?
3. The performance is not very promising
4. Why not implement 5-fold cross validation on AMOS 2022? It has 300 CT cases, why only use 200 cases here?

---

### Official Review · Reviewer_i9HZ · 2024-11-03

**Soundness:** 2
**Presentation:** 1
**Contribution:** 3
**Rating:** 5
**Confidence:** 4

**Summary:**

In this paper, the authors propose an efficient novel learning framework that captures token relations through progressive summarization of features. For that, they use the compaction capability of discrete wavelet transform (DWT) on high-dimensional features. The authors evaluated the framework on three publicly available datasets, where it achieves state-of-the-art performance.

**Strengths:**

The paper proposes an efficient architecture for medical image segmentation tasks.

The authors performed experiments on challenging 3D volumetric segmentation benchmarks and provided detailed hyperparameters to reproduce their model’s performance.

**Weaknesses:**

Missing recent state-of-the-art models during the comparison, e.g., Auto3DSeg and MedNeXt

Though the paper delivers a novel idea, it is disorganised. For example, figure references are missing in various places (Line 421, 430), and figures and captions are not appropriately aligned (Figure 4).

**Questions:**

The paper uses a fixed window size for attention, determined by the spatial dimension of the coarsest-level feature. How were this specific window size chosen?

The authors mentioned that “WaveFormer captures local and global relations in each layer on the multiscale low-frequency approximations obtained using DWT.” Providing activation or attention maps could be very helpful in understanding how WaveFormer captures local and global relationships using DWT.

In Table 6, the decomposition levels for each stage are presented for different WaveFormer configurations. It's unclear what "0" means. Does it imply that no DWT is applied at that stage? How does this impact the overall performance and the capture of multi-scale features?

---

### Official Review · Reviewer_SatB · 2024-11-04

**Soundness:** 3
**Presentation:** 2
**Contribution:** 2
**Rating:** 3
**Confidence:** 4

**Summary:**

This paper introduces WaveFormer, a medical image segmentation model that utilizes discrete wavelet transform (DWT) to compact features and capture long-range dependencies across multiple scales. By leveraging DWT, WaveFormer efficiently represents fine-grained local and coarse global contextual information, while reducing computational complexity for self-attention on transformed features. WaveFormer is particularly suited to handling volumetric medical image data. Experimental results on three challenging medical segmentation tasks demonstrate the efficiency and effectiveness of WaveFormer.

**Strengths:**

1. The paper is well-structured, with a clear and sound motivation. It addresses one of the main bottlenecks of applying transformers to volumetric data, high complexity, by approaching it from a signal processing perspective.

2. Beyond medical image segmentation, the authors also assess classification performance on ImageNet-1K. Although the improvement is modest, WaveFormer achieves comparable accuracy with reduced computational demands.

**Weaknesses:**

1. Some references are missing (e.g., Lines 421 and 430).

2. Captions and figures overlap (e.g., Line 403).

3. The overall improvement in segmentation accuracy is marginal, with roughly a 1% Dice increase over SwinUNETR on the FLARE 2021 dataset, while computational complexity remains similar. A statistical significance test is recommended to confirm if the improvement is meaningful.

4. GPU memory footprint comparisons are absent; reductions in FLOPs and parameters alone appear marginal.

5. The paper appears incomplete.

**Questions:**

1. Where are the results for the MICCAI 2019 KiTS Challenge?

2. Did the authors use nnU-Net V1 or V2?

---

### Official Review · Reviewer_y7nE · 2024-11-04

**Soundness:** 2
**Presentation:** 1
**Contribution:** 3
**Rating:** 3
**Confidence:** 4

**Summary:**

The paper proposes a new transformer architecture based on the discrete wavelet transform to learn multi-scale token interactions. The architecture is proposed for volumetric data, and evaluated on three segmentation tasks on 3D CT scans. The classification performance of a 2D version of the transformer is evaluated on imagenet-1k.

**Strengths:**

* **(S1)** The application of Discrete Wavelet Transform (DWT) to volumetric medical data presents a compelling and innovative approach.
* **(S2)** The proposed methodology is both novel and makes a contribution to existing approaches, particularly in its application to volumetric medical data.
* **(S3)** The method demonstrates versatility and can be applied broadly to various types of volumetric data.
* **(S4)** The introductory sections are clearly articulated and well-structured, providing a good foundation for the paper.
* **(S5)** The evaluation includes comprehensive comparisons with state-of-the-art models on biomedical data, notably with the nnUNet model.

**Weaknesses:**

In short, the paper does not substantiate its claims, lacks thorough analysis to support its methodology, and suffers from a general lack of clarity.

- **(W1)** The title of the paper is phrased as a general methodology for both 2d and volumetric data, however the methodology is throughout build as an evolution of the UNETR architecture for volumetric medical data and all figures center around 3D input. The application to 2D data is non-trivial, and not explained including in the appendix.
- **(W2)** Experimental results on Imagenet-1k does not meaningfully beat prior methods, and the authors does not provide any analysis to justify the applicability of the Waveformer methodology for this data.
- **(W3)** The paper claims to be evaluated on the KiTS dataset (mentioned both in abstract and in section 5), however the results on this dataset are not given.
- **(W4)** The paper generally lacks in-depth analysis of the effects and implications of performing attention on DWT coefficients. In particular, the claims made on lines 205-215 would be very relevant for empirical analysis, however this is missing from the paper.
- **(W5)** While the authors claim that the performance boost of the methodology is caused by enlarged per layer receptive field, the two MICCAI challenge datasets used are generally characterised by requiring only local information (e.g. intra organ features) to perform, and the authors provide no explanation for this.
- **(W6)** In general, the paper lacks clarity. The writing is overly complicated and the figures of the paper are generally neither well-made nor easily digestible. In particular Figure 2, which is of major importance to understand the paper, is difficult to understand. Further, Figure 3 is almost a copy of the figure provided in SwinUNETR [1]. Lastly, the figure text is on top of Figure 4.
- **(W7)** Paper lacks central details on the DWT implementation, e.g. choice of kernel.

**Questions:**

General:
* Can you provide evaluation results on KiTS?
* The paper [2] generally questions the evaluation of SwinUNETR and similar transformer based methods in medical image segmentation. How did the authors overcome the problems highlighted in this paper and did they conform to the proposed recommendations?

DWT:

* What is the effect of multiple applications of DWT and what is intuition about why this is effective?
* What kernel is used?

Technical questions:

* When applying transformers to volumetric data we have two notions of patches: A patch extracted from the volume of size 96^3 which is then further divided into patches to be embedded before being processed by the transformer. What is the patch size used for the transformer?
* The SwinUNETR methodology used ensembles, post-processing of the results and test-time augmentations [1] to be competitive with nnUnet. Did the authors use any of these to obtain the displayed results?

[1] https://github.com/Project-MONAI/research-contributions/issues/143

[2] Isensee, Fabian, Tassilo Wald, Constantin Ulrich, Michael Baumgartner, Saikat Roy, Klaus H. Maier-Hein and Paul F. Jaeger. “nnU-Net Revisited: A Call for Rigorous Validation in 3D Medical Image Segmentation.” MICCAI 2024 (2024): n. pag.

---

### Note · Authors · 2024-11-25

**Comment:**

Greetings,
Thank you for the valuable and detailed reviews. Based on the provided suggestions, we are thinking of enhancing our work and will try to submit a complete version in a future venue.

Thank you again.

Best,
Authors

**Withdrawal Confirmation:**

I have read and agree with the venue's withdrawal policy on behalf of myself and my co-authors.